# ARSS: TAMING DECODER-ONLY AUTOREGRESSIVE VISUAL GENERATION FOR VIEW SYNTHESIS FROM SINGLE VIEW

## ABSTRACT

Diffusion models have achieved impressive results in world modeling tasks, including novel view generation from sparse inputs. However, most existing diffusion-based NVS methods generate target views jointly via an iterative denoising process, which makes it less straightforward to impose a strictly causal structure along a camera trajectory. In contrast, autoregressive (AR) models operate in a causal fashion, generating each token based on all previously generated tokens. In this work, we introduce **ARSS**, a novel framework that leverages a GPT-style decoder-only AR model to generate novel views from a single image, conditioned on a predefined camera trajectory. We employ a video tokenizer to map continuous image sequences into discrete tokens and propose a camera encoder that converts camera trajectories into 3D positional guidance. Then to enhance generation quality while preserving the autoregressive structure, we propose a autoregressive transformer module that randomly permutes the spatial order of tokens while maintaining their temporal order. Qualitative and quantitative experiments on public datasets demonstrate that our method achieves overall comparable to state-of-the-art view synthesis approaches based on diffusion models. Our code will be released upon paper acceptance.

## 1 INTRODUCTION

World models (Huang et al., 2024; Zheng et al., 2024) are internal, learned representations in AI systems that simulate the real world, allowing agents to understand, predict, and plan future events by modeling physical dynamics or spatial relationships. One important application of world models is to explore and construct a 3D space given very sparse initial inputs. This task requires the system to generate high-quality, content-consistent novel views from an input image and a pre-defined camera trajectory. To scale to large environments and long trajectories, it is desirable to process observations in a sequential and causal manner, synthesizing new views conditioned on both the inputs and previously accumulated generations. Recent advances in diffusion models have significantly boosted the performance of novel view synthesis from sparse or even single inputs (Ren et al., 2025b; Cao et al., 2025; Yu et al., 2024b; Zhou et al., 2025). However, many of these diffusion-based NVS methods generate target views jointly via iterative denoising in a high-dimensional latent space, which can make it less straightforward to impose a strictly causal structure along a camera path or to incrementally extend and reuse existing generations when the trajectory changes.

The advent of autoregressive (AR) models in visual generation task (Esser et al., 2021; Sun et al., 2024; Yu et al., 2024a; Pang et al., 2025; Tian et al., 2024; Wang et al., 2025) has shown promising results from modeling image synthesis as a sequential and causal process. These methods first utilize an image tokenizer to encode images into discrete tokens and then apply a GPT-style causal transformer for next-token prediction. While these methods demonstrate the feasibility of AR model in single-image visual generation, their applications in novel view synthesis have hardly been explored, as generating novel views require precise camera control and 3D spatial awareness. Inspired by these works, we believe that AR models have the potential as novel view synthesizer for world models that require construction of a large 3D space.

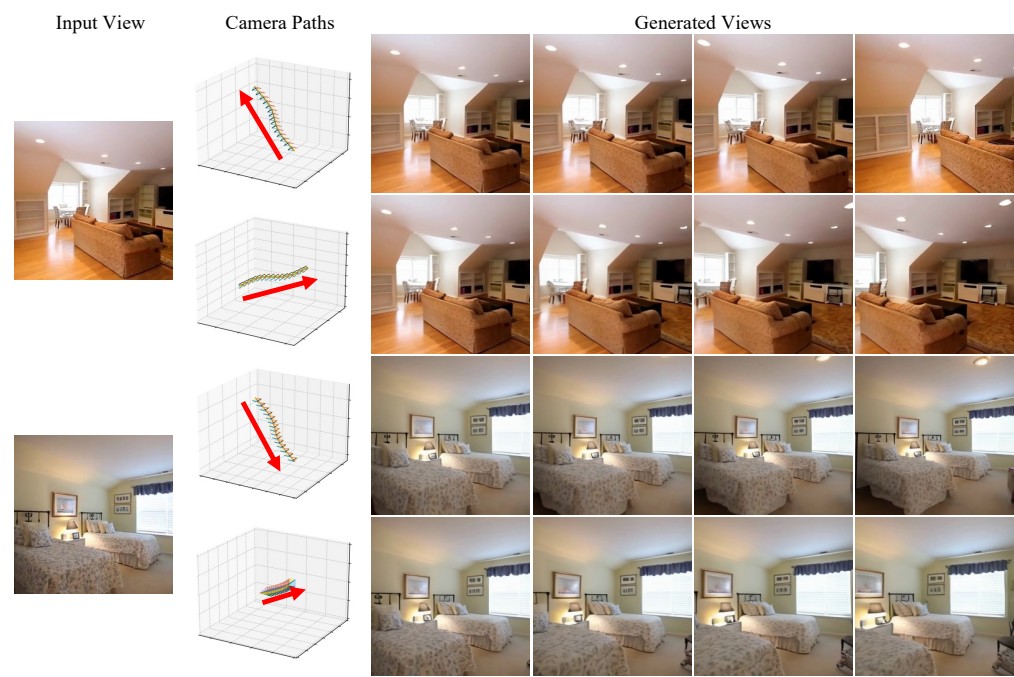

Figure 1: **Illustration of ARSS.** Given a source input image and camera trajectory, ARSS can generate photorealistic and 3D consistent novel views. Although a lot of previous methods tackle the same task with other generative model like diffusion models (Rombach et al., 2022; Ho et al., 2022), ARSS is the first that leverages decoder-only causal transformer and generate multi-views with a next-token prediction style.

Therefore, in this work, we propose **A**uto**R**regressive Novel View **S**ynthesis from a **S**ingle Image (ARSS), a novel approach that applies the causal decoder-only transformers to generate novel views from a single image conditioned on a pre-defined camera trajectory. We aim at nesting sequential view generation into a next-token-prediction norm while preserving 3D spatial awareness. The process involves tokenizing the multi-view image sequences into discrete codes and maximize the likelihood of the current token given all previous tokens with an autoregressive transformer. However, the AR image generation pipeline has the following three problems: First, previous autoregressive visual generation relies on Vector-Quantization (VQ) (Esser et al., 2021) for image tokenization. However, temporal consistency is hard to preserve if VQ is applied for independent per-frame tokenization. To address this issue, we adopt a video tokenizer (Tang et al., 2024), which incorporates both spatial and temporal encoding, to convert the multi-view image sequences into compact latent tokens.

Second, current autoregressive visual generation usually prefills the output sequence with a class token for conditional generation. However, it is hard to encode the camera trajectory into a global token to guide sequence generation. Therefore, we propose to pair each discrete tokens with a 3D positional guidance token extracted from the pre-defined camera trajectory. To achieve this goal, we design a camera autoencoder that maps the Plücker raymap (Plucker, 1865) to camera latent features, which possess the same spatial and temporal dimension as the visual latent tokens. We pre-train the camera autoencoder such that the encoded camera features contain the information of original camera trajectory.

Third, visual data are semantically low-level and present bi-directional context. Directly training a uni-directional causal transformer on bi-directional 2D images may lead to suboptimal solutions (Li et al., 2024a). Inspired by previous works (Pang et al., 2025; Yu et al., 2024a), we propose to randomly shuffle the spatial orders of visual tokens such that uni-directional transformer would be optimized with all the permutations of bi-directional data. While training with random spatial permutation of image tokens, the temporal order is still maintained to make sure that the tokens from later frames are always generated based on tokens of former frames. According to Pang et al. (2025) and Yu et al. (2024a), positional instruction tokens are the key factor of the randomly shuffled image

tokens. As a matter of fact, the proposed camera tokens indeed provide accurate 3D position in the scene. Therefore, when predicting the next visual token, a camera token can be inserted, representing the 3D positional information of the current token to be generated. Through the aforementioned modules, ARSS integrates novel view synthesis with autoregressive training and sampling paradigm as well as achieving precise camera control. To the best of our knowledge, ARSS is the first that applies the GPT-style causal autoregressive model in novel view generation with camera control.

We train and evaluate the proposed pipeline on public datasets including RealEstate-10K (Zhou et al., 2018) and ACID (Liu et al., 2021). To demonstrate the generalization capability of ARSS, we conduct zero-shot novel view synthesis experiment on DL3DV benchmark (Ling et al., 2024). Both qualitative and quantitative results demonstrate that our method out-performs current state-of-the-art methods.

## 2 RELATED WORKS

**Novel View Generation with Diffusion Models** Diffusion models (Rombach et al., 2022; Song et al., 2020; Meng et al., 2021; Yin et al., 2025) learn a desnoising process that maps a Gaussian noise to clean samples conditioned on class labels, text prompts, etc. Leveraging diffusion models for novel view synthesis (Zhou et al., 2025; Yu et al., 2024b; Ren et al., 2025b; Cao et al., 2025; Gao et al., 2024; Wu et al., 2024; Watson et al., 2024; Chen et al., 2024; Liu et al., 2024; Wu et al., 2025) is to generate the target novel view given arbitrary number of source input views and both source and target camera poses. Some of these methods (Yu et al., 2024b; Ren et al., 2025b; Liu et al., 2024; Chen et al., 2024) construct 3D prior from source inputs and provide globally per-view condition to the diffusion model. Majority of these methods apply a video diffusion model to revise the 3D inductive bias. Some other methods (Gao et al., 2024; Cao et al., 2025; Zhou et al., 2025) use binary mask to differentiate between source and target views and apply multi-view diffusion model to directly generate the target views. Although these methods excel at generating photo-realistic images or sequences, they have to generate all the images simultaneously, which is hard to adapt to new input or generate based on accumulated knowledge.

**Autoregressive Visual Generation** Autoregressive model applies a causal model (e.g. GPT model (Brown et al., 2020)) to generate samples sequentially based on previous accumulated information, and has seen promising application in language modeling tasks (Achiam et al., 2023; Bai et al., 2023; Chowdhery et al., 2023; Grattafiori et al., 2024; Radford et al., 2018; Team et al., 2023; Touvron et al., 2023). Some current researches (Esser et al., 2021; Sun et al., 2024; Yu et al., 2024a; Pang et al., 2025; Tian et al., 2024; Wang et al., 2025; Li et al., 2024b; Huang et al., 2025; Ren et al., 2025a) focus on integrating autoregressive model into visual generation task. LlamaGen (Sun et al., 2024) is the pioneer work for autoregressive visual generation but the generation is required to follow a raster-scan order, which is incompatible with bi-directional data structure of image data. Recent works (Pang et al., 2025; Yu et al., 2024a; Wang et al., 2025) purpose to reorder the image tokens to adapt the uni-directional model. Both Pang et al. (2025) and Yu et al. (2024a) purpose to randomly shuffle the image tokens and insert positional instruction tokens for positional guidance. Wang et al. (2025) designs a novel approach that divide image tokens into sections and generate tokens at different sections simultaneously. However, these methods focus only on image generation and none of the methods focus on video or multi-view sequence generation.

**Video Tokenization** Most of the video tokenization methods adopt an encoder-decoder architecture. The encoder will compress the video data into latent tokens w.r.t. both spatial and temporal dimensions, whereas the decoder reconstructs the latent tokens back to pixels. Vector Quantized-Variational Autoencoder (VQ-VAE) (Van Den Oord et al., 2017) is introduced to map the encoded features into a finite set of vectors in a codebook. By contrast, Tang et al. (2024) proposed to apply Finite Scalar Quantization (FSQ) (Mentzer et al., 2023) to obtain discrete tokens. Different from Vector Quantization (VQ), FSQ releases from learning the large codebook, thus stabilizes and facilitates training.

## 3 METHOD

### 3.1 PRELIMINARY

**Autoregressive Visual Generation.** Given a discrete 1-D token sequence, denoted as $x = [x_1, x_2, ..., x_N]$, an autoregressive model is trained to maximize the probability of each token $x_i$

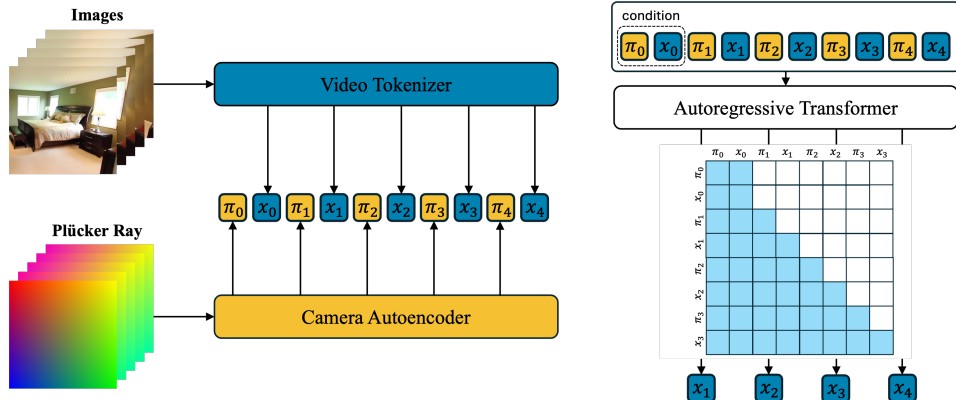

Figure 2: **Overall architecture of our proposed method.** Left: we apply a video tokenizer to convert image sequence into latent codes. We also apply a camera autoencoder to map camera Plücker raymap to latent camera tokens. The camera tokens are inserted before visual tokens as a 3D positional instruction. Right: the interleaved sequence is the input of a decoder-only causal transformer. The tokens of the first view are the condition tokens thus always visible to all the subsequent tokens. We use the ground truth sequence from the tokenization process to supervise the weights of autoregression model.

given all the previous tokens:

$$\max_\theta p_\theta(\boldsymbol{x}) = \prod_{i=1}^{N} p(x_i|x_1, x_2, ..., x_{i-1}) = \prod_{i=1}^{N} p(x_i|x_{<i}), \tag{1}$$

where $p_\theta$ is a probability predictor parameterized by $\theta$. Given the background of image generation, $x_i$ in Eq. 1 represents the image tokens usually obtained by vector quantization in previous works (Sun et al., 2024; Esser et al., 2021) and the total number of tokens in the sequence ($N$) equals to the number of image tokens in latent space ($N = h \times w$, where $h$ and $w$ are the compressed dimensions in $y$, $x$ coordinates, respectively). In addition, previous visual generation method would add a class label embedding or text embedding $c$ as a condition at the start of the sequence, and the image generation process is further formulated as:

$$\max_\theta p_\theta(\boldsymbol{x}) = \prod_{i=1}^{h \times w} p(x_i|x_{<i}, c). \tag{2}$$

Concurrent works(Esser et al., 2021; Sun et al., 2024; Yu et al., 2024a; Pang et al., 2025; Tian et al., 2024; Wang et al., 2025) use a causal decoder-only transformer to model the sequence by minimizing the following optimization function:

$$\mathcal{L} = CE(f_\theta([c, x_1, x_2, ..., x_{N-1}]), [x_1, x_2, ..., x_N]), \tag{3}$$

where $CE$ stands for the cross entropy loss, $f$ is the decoder-only transformer with $\theta$ the trainable parameters.

**Causal Video Tokenization.** Similar to VQ-VAE for image tokenizer, video tokenizer consists of an encoder $\boldsymbol{E}$, a decoder $\boldsymbol{D}$ and a regularizer $\boldsymbol{R}$. Given a video sequence $\boldsymbol{X} \in \mathbb{R}^{L \times 3 \times H \times W}$, the encoder $\boldsymbol{E}$ compresses $\boldsymbol{X}$ into latent space and decoder $\boldsymbol{D}$ reconstructs the latent feature back to original:

$$\boldsymbol{Z} = \boldsymbol{R}(\boldsymbol{E}(\boldsymbol{X})), \hat{\boldsymbol{X}} = \boldsymbol{D}(\boldsymbol{Z}), \tag{4}$$

where $\boldsymbol{Z} \in \mathbb{R}^{l \times 3 \times h \times w}$ is the latent representation. For non-causal scenario, $L = r_t \times l$ and $H = r_s \times h, W = r_s \times w$, where $r_t$ and $r_s$ are the temporal and spatial compression ratio, respectively. For causal scenario, the original video input is $\boldsymbol{X} \in \mathbb{R}^{(L+1) \times 3 \times H \times W}$, which contains $L + 1$ frames

will be compressed into $\boldsymbol{Z} \in \mathbb{R}^{(l+1)\times 3 \times h \times w}$. The first frame is independent from the subsequent frames and will not be compressed along temporal dimension. We use the token of the first frame as the conditional tokens filled at the start of the result sequence.

## 3.2 ARSS FRAMEWORK

**Overview.** In this section, we will introduce our proposed method, ARSS, with overall workflow depicted in Figure 2. ARSS performs novel view synthesis from a single image input, by relying on a transformer that performs next-token prediction. Different from the typical autoregressive transformer models (e.g. VQGAN) that synthesize a single image, ARSS needs to autoregressively predict visual tokens that are 1) coherent in a sequence/multiple views, 2) controllable by camera trajectories and 3) generated in a manner that captures long-range dependencies across tokens. The formation of our method is therefore driven by three important modules: a **video tokenizer** that converts multi-view images into compact visual tokens while preserving temporal consistency, a **camera autoencoder** that encodes camera trajectories into camera tokens serving as positional guidance, and an **autoregressive transformer module** that predicts the next token conditioned on both previously generated visual tokens and the corresponding camera tokens.

### 3.2.1 LEARNING VISUAL TOKENS FOR NOVEL VIEW SYNTHESIS

The main challenge in novel view synthesis lies in modeling both visual details and temporal consistency across multiple frames. An image tokenizer fails to capture inter-frame relationships, often lead to temporal artifacts (e.g flickering). To address this, we adopt a video tokenizer that compress multi-view sequences into tokens while preserving temporal structure. By preserving temporal dependencies, the video tokenizer provides a more robust representation for novel view synthesis, leading to improved consistency and quality as demonstrated in Section 4.3.

Formally, given an original multi-view image sequence $\boldsymbol{X} \in \mathbb{R}^{(L+1)\times C \times H \times W}$ and the corresponding camera poses $\boldsymbol{\Pi} \in \mathbb{R}^{(L+1)\times 6 \times H \times W}$, where $L$ denotes the temporal length and $H, W$ the spatial dimensions, we process $\boldsymbol{X}$ using a video tokenizer (Tang et al., 2024). The tokenizer learns to encode $\boldsymbol{X}$ into a sequence of one-dimensional discrete tokens $[x_1, x_2, \ldots, x_N]$, where the sequence length is $N = l \times h \times w$ and $(l, h, w)$ correspond to the compressed temporal and spatial dimensions.

### 3.2.2 LEARNING 3D POSITIONAL TOKENS

While the video tokenizer provides temporally consistent visual tokens, novel view synthesis also requires 3D guidance to ensure that generated views align with the underlying camera trajectory. To address this, we explicitly incorporate 3D geometry by converting the per-frame extrinsic and intrinsic matrices into Plücker raymaps $\boldsymbol{\Pi}$. These raymaps are then compressed into a sequence of camera tokens $[\pi_1, \pi_2, \ldots, \pi_N]$ using a dedicated camera autoencoder. The trajectory thereby provides direct global 3D structural information for multi-view sequence. The autoencoder follows a conventional encoder–decoder design: the encoder maps Plücker coordinates into a latent representation using stacked 3D convolutional and downsampling blocks, while the decoder reconstructs them with symmetric 3D convolutional and upsampling blocks. Different from image or video autoencoder that applies reconstruction loss, perpetual loss and adversarial loss, we add geometry constraints to enforce geometry consistency:

$$\mathcal{L}_{\text{cam}} = \lambda_1 \|\hat{\boldsymbol{d}} - \boldsymbol{d}\|_2^2 + \lambda_2 \|\hat{\boldsymbol{m}} - \boldsymbol{m}\|_2^2 + \lambda_3 (\|\hat{\boldsymbol{d}}\| - 1)^2 + \lambda_4 (\hat{\boldsymbol{d}} \cdot \hat{\boldsymbol{m}})^2, \tag{5}$$

where $\boldsymbol{d}$ is the normalized camera ray direction, $\boldsymbol{d}$ is the momentum term formulated as $\boldsymbol{m} = \boldsymbol{o} \times \boldsymbol{d}$. The first two loss terms are l2-norm reconstruction loss. The third term regularizes the camera rays have unit length. The last term regularizes that the camera rays $\boldsymbol{d}$ and momentum $\boldsymbol{m}$ are orthogonal.

### 3.2.3 NEXT-TOKEN PREDICTION FOR NOVEL VIEW SYNTHESIS

With visual tokens that capture appearance and temporal consistency, and camera tokens that encode explicit 3D geometry, the final step is to synthesize novel views through token prediction. To this end, we designed an autoregressive transformer module that performs next token prediction. This design is made effective by two key components: (1) a hybrid token order permutation strategy that preserves temporal causality while enabling the model to exploit bi-directional spatial context, and

(2) a training objective that aligns the autoregressive prediction with this ordering to improve both fidelity and temporal consistency.

**Token order permutations.**    Previous autoregressive transformers employ causal attention masks, which impose a strict uni-directional dependency across the token sequence. This is misaligned with visual data, where spatial context within each frame is inherently bi-directional. To address this, we introduce a hybrid ordering strategy. Specifically, we permute the spatial order of tokens within each frame while preserving the original temporal order across frames. This permutation strategy would guarantee that the tokens from views far from the input would be generated after those close to the input. The permuted sequence $\mathcal{S}$ can be illustrated as the following:

$$\mathcal{S} = [\pi_{11}^{P_1(1)}, x_{11}^{P_1(1)}, ..., \pi_{1n}^{P_1(n)}, x_{1n}^{P_1(n)}, \pi_{21}^{P_2(1)}, x_{21}^{P_2(1)}, ..., \pi_{2n}^{P_2(n)}, x_{2n}^{P_2(n)}, ..., \pi_{ln}^{P_l(n)}, x_{ln}^{P_l(n)}], \quad (6)$$

where $x_{ij}^{P_i(j)}$ represents the $j$-th randomly shuffled token under $i$-th frame, where $i \in \{1, 2, .., l\}$ and $j \in \{1, 2, ..., n\}$. This means any given token $x_{ij}$ can only be swapped with $x_{ik}$, where $1 \leq j \neq k \leq n$.

**Training objective and sampling.**    The shuffled tokens in Eq. 6 are fed into a decoder-only transformer for next-token prediction. During optimization, the final objective function (Eq. 3) can be re-formulated as:

$$\mathcal{L} = CE(f_\theta([\mathcal{S}, [x_{21}^{P_2(1)}, ..., x_{ln}^{P_l(n)}]), \quad (7)$$

Given that the first frame is the input, so the corresponding visual and camera tokens are always visible to the subsequent tokens. During generation, the autoregressive model (Eq. 2) can be re-formulated as:

$$\max_\theta p_\theta(\boldsymbol{x}) = \prod_{i=2}^{l} \prod_{j=1}^{n} p(x_{ij}^{P_i(j)} | \pi_{\leq i, \leq j}^{P_{\leq i}(\leq j)}, x_{<i, <j}^{P_{<i}(<j)}, [\pi_{11}^{P_1(1)}, x_{11}^{P_1(1)}, ..., \pi_{1n}^{P_1(n)}, x_{1n}^{P_1(n)}]) \quad (8)$$

where $\pi_{\leq i, \leq j}^{P_{\leq i}(\leq j)}$ contains the camera tokens for the current and previously generated tokens denoted as $x_{<i, <j}^{P_{<i}(<j)}$. $[\pi_{11}^{P_1(1)}, x_{11}^{P_1(1)}, ..., \pi_{1n}^{P_1(n)}, x_{1n}^{P_1(n)}]$ are the input tokens prefilled before the output sequence. Another advantage of randomly shuffle tokens is that it allows parallel decoding (Pang et al., 2025). The generation of current token doesn't need to rely on the tokens spatially surrounding it. With camera tokens as positional instruction tokens, the system has the capacity to predict multiple tokens at one time.

## 4 EXPERIMENTS

### 4.1 EXPERIMENTS SETUP

**Datasets.**    We use the RealEstate10K dataset (Zhou et al., 2018) and ACID dataset (Liu et al., 2021) to train and validate our proposed method. RealEstate10K is a large dataset with over 80K indoor and outdoor scenes from over 10K YouTube videos. ACID is dataset of aerial footage of natural coastal scenes. To further validate our method, we also evaluated our proposed method on the benchmark set of DL3DV-10K (Ling et al., 2024) dataset for zero-shot novel view synthesis. DL3DV is a large-scale scene dataset comprising both bounded and unbounded scenes from over 10L videos.

**Baselines and Evaluation Metrics.**    We compare ARSS against both non-diffusion and diffusion-based baselines for novel view and video generation. As non-diffusion NVS methods, we include LVSM (Jin et al., 2024), a transformer-based architecture for sparse-view novel view synthesis, and RayZer (Ren et al., 2025b), which leverages explicit 3D-aware representations for multi-view consistent rendering. Among diffusion-based approaches, we consider SEVA (Zhou et al., 2025) (multi-view diffusion for NVS), Genwarp (Seo et al., 2024) (warping-guided diffusion), MotionCtrl (Wang et al., 2023) (controls camera and object motion), and ViewCrafter (Yu et al., 2024b).

| Input | SEVA | Genwarp | LVSM | Ours | Ground Truth |

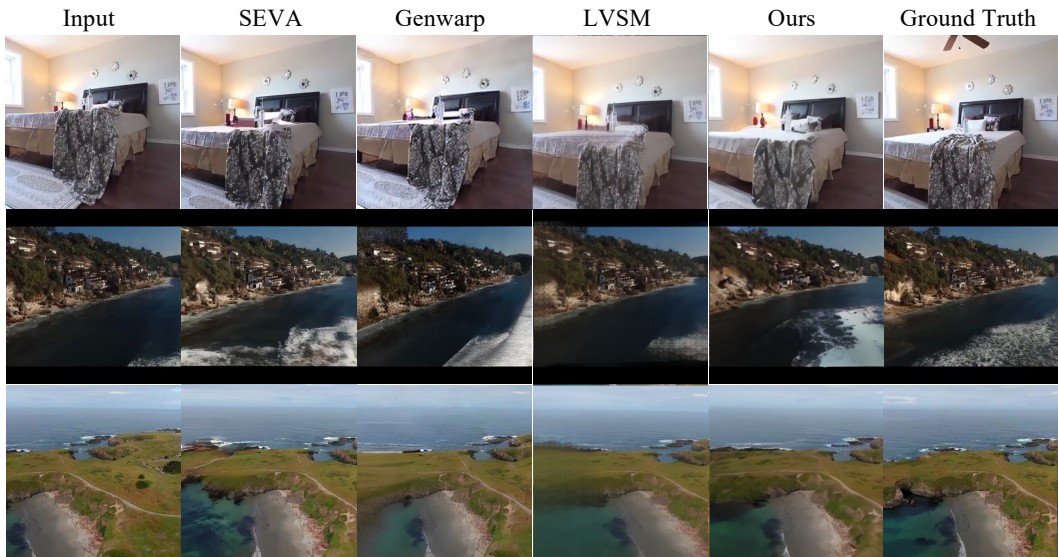

Figure 3: **Qualitative Visualization.** Qualitative comparison between ARSS with other diffusion-based and feed-forward transformer-based methods on ReaEstate10K and ACID datasets. Diffusion-based methods such as SEVA and Genwarp often suffer from distortions and inaccurate camera pose alignment, while the feed-forward transformer-based LVSM produces results that are noticeably blurry along boundaries. In contrast, ARSS generates geometrically consistent and sharp views across diverse scenes.

Table 1: **Quantitative results on RealEstate10K, ACID, and DL3DV**. Higher PSNR/SSIM and lower LPIPS/FID/FVD are better. For SEVA, ViewCrafter and RayZer results on DL3DV are not reported, since DL3DV was part of its training data, while for other methods it serves as zero-shot evaluation. We highlight the best results in red and second-best in yellow.

| | Re10K | | | | | ACID | | | | | DL3DV | | | | |
| Method | PSNR ↑ | SSIM ↑ | LPIPS ↓ | FID ↓ | FVD ↓ | PSNR ↑ | SSIM ↑ | LPIPS ↓ | FID ↓ | FVD ↓ | PSNR ↑ | SSIM ↑ | LPIPS ↓ | FID ↓ | FVD ↓ |
|---|---|---|---|---|---|---|---|---|---|---|---|---|---|---|---|
| MotionCtrl | 16.17 | 0.609 | 0.438 | 59.73 | 63.58 | 19.36 | 0.626 | 0.405 | 63.93 | 66.30 | 14.58 | 0.430 | 0.507 | 92.91 | 94.85 |
| ViewCrafter | 12.67 | 0.399 | 0.490 | 121.25 | 108.99 | 16.96 | 0.504 | 0.442 | 102.48 | 104.85 | - | - | - | - | - |
| RayZer | 12.97 | 0.397 | 0.639 | 324.23 | 130.23 | 12.64 | 0.384 | 0.6521 | 303.75 | 138.62 | - | - | - | - | - |
| LVSM | 18.29 | 0.579 | 0.314 | 50.29 | 56.31 | 20.81 | 0.573 | 0.308 | 38.46 | 55.13 | 15.86 | 0.409 | 0.400 | 85.75 | 96.83 |
| SEVA | 18.73 | 0.670 | 0.349 | 46.98 | 57.56 | 21.77 | 0.664 | 0.326 | 33.16 | 53.69 | - | - | - | - | - |
| Ours | 19.02 | 0.624 | 0.269 | 47.60 | 50.51 | 21.93 | 0.623 | 0.265 | 47.76 | 54.60 | 16.70 | 0.449 | 0.347 | 84.96 | 91.25 |

We evaluate all methods using pixel-aligned metrics (PSNR, SSIM (Wang et al., 2004)), perceptual metrics (LPIPS (Zhang et al., 2018)), and distributional video/image metrics (FID (Heusel et al., 2017) and FVD (Unterthiner et al., 2019)).

**Implementation Details.** For the decoder-only transformer, we adopt LlamaGen (Sun et al., 2024) as our backbone model and the dimension is set to be 1280. We train ARSS with 8 NVIDIA H100 GPUs with a batch size of 8 per GPU for 100K interations. The learning rate is set to $5e-4$ with 5K steps warm up and a cosine schedule to decrease to 0 after the warm up steps. . We apply VidTok (Tang et al., 2024) as our video tokenizer for temporally causal modeling. The spatial patch size is 8 and temporal patch size is 4. All the images are in a resolution $256 \times 256$ and the temporal dimension is 17, so the video tokenizer will extract $17 \times 256 \times 256$ image sequence into $5 \times 32 \times 32$ latent codes. The first $32 \times 32$ tokens are the input tokens and their orders would not be permuted. During inference, we prefill the camera tokens and the visual tokens of the input view as well as the camera tokens of the first target views to the sequence, and iteratively sample the target tokens using a next-token prediction manner.

| Input | MotionCtrl | Genwarp | LVSM | Ours | Ground Truth |

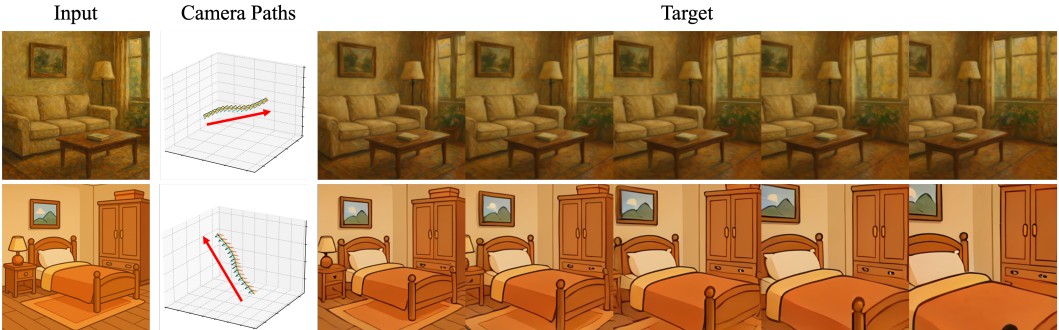

Figure 4: **Qualitative Visualization.** Zero-shot novel view synthesis comparison between ARSS with other diffusion-based and feed-forward transformer-based methods on DL3DV benchmark (Ling et al., 2024). MotionCtrl and Genwarp exhibit distortions due to incorrect camera pose alignment, while LVSM produces results that are noticeably blurry. Our proposed method, ARSS, generates sharp views with geometric consistency

| Input | Camera Paths | | Target |

Figure 5: **View Generation Results.** Zero-shot novel view synthesis visualization on AI Generated Betker et al. (2023) images. The results demonstrate the strong generalizability of our method, generating consistent and high-fidelity novel views even when applied to out-of-distribution, synthetically generated inputs.

## 4.2 RESULTS

**Qualitative Results.** We provide qualitative comparison between our proposed method and three baseline methods in Figure 3. Our method visually outperforms majority of the baseline methods for in-domain testing, demonstrating the strong capability of generating both photorealistic and geometrically consistent novel views from a single image. Genwarp (Seo et al., 2024) follows a warp-and-inpaint paradigm and highly rely on the metric accuracy of predicted depth and camera transitions, thus may generate samples with erroneous camera poses or apparent artifacts. LVSM (Jin et al., 2024) applies bi-directional transformer to directly predict visual tokens thus cannot generate views based on previous knowledge. SEVA tends to generate high quality and 3D consistent novel views, but it follows a paradigm that first generates anchor views and interpolate the intermediate views between input and anchor views, which may sometimes cause content and view inconsistency.

**Quantitative Results.** We present quantitative comparisons in Table 1. Our method consistently outperforms most of the baselines: Genwarp and MotionCtrl (Wang et al., 2023) underperform across metrics due to the lack of explicit modeling of relative camera poses, showing stability only for nearby views but degrading with larger viewpoint changes, while LVSM, which relies on feed-forward predictions rather than generative modeling, resulting poor performance. SEVA (Zhou et al., 2025) achieves results relatively close to ours, but although our method produces higher-fidelity novel views (e.g., +1.1% PSNR, −21% LPIPS), it can show minor geometric inconsistencies (e.g., −6.6% SSIM, +22% FID). It is worth noting that SEVA benefits from large-scale, high-resolution

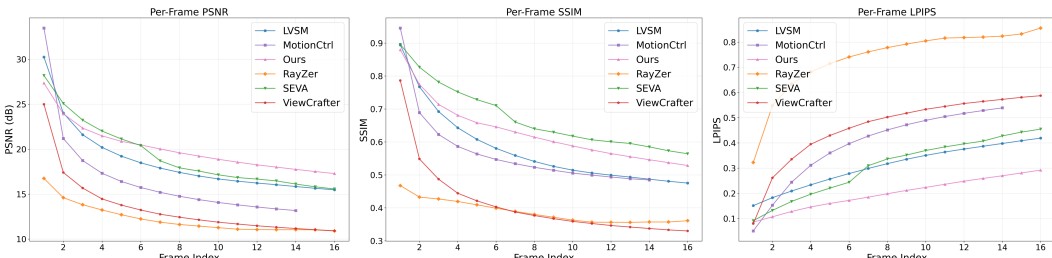

Figure 6: **Error accumulation analysis.** Per-frame PSNR/SSIM/LPIPS vs. frame index, showing that our method maintains consistently higher image quality and slower degradation than baseline methods along camera trajectories.

training data and heavy computational resources, whereas our approach attains competitive performance without such requirements.

**Zero-shot Novel View Synthesis.** We directly validate our proposed method on DL3DV benchmark (Ling et al., 2024) and compare with other state-of-the-arts method. MotionCtrl (Wang et al., 2023) is capable of generating images with richer and sharper details but fail to model the relative camera positioning between source and target views. Both Genwarp (Seo et al., 2024) and LVSM (Jin et al., 2024) exhibit apparent artifacts and geometry bias of target views. Compared to baseline methods, our method can generate both 2D and 3D consistent novel views. In addition, Figure 5 show qualitative result on AI-generated (Betker et al., 2023) oil and cartoonish pictures. The results demonstrate the strong generalizability of our method, consistently producing high-quality novel views from diverse input image styles under predefined camera trajectories.

**Error Accumulation Analysis.** Visualized in Figure 6, our method shows clearly better long-horizon behavior than all baselines. As the frame index increases, our model maintains consistently highest or near-highest PSNR/SSIM while exhibiting the lowest LPIPS at every timestep, indicating both strong pixel-level accuracy and superior perceptual fidelity. Moreover, the slopes of all three curves for our method are noticeably flatter, meaning quality degrades much more slowly along the trajectory. Taken together, these per-frame metrics demonstrate that our approach accumulates significantly less error over time and is overall superior to competing methods for long camera sweeps.

## 4.3 ABLATION STUDIES

**Ablation on token order permutation.** We first compare different ways to permute the target tokens and the results are shown in Figure 7. One permutation strategy is to keep the original token order as it is not permuted, where tokens are ordered from top left to bottom right spatially and from the first to the last view temporally. We notate it as "raster" order and the results are shown in the third row of Figure 7. Another permutation strategy is to randomly shuffle all the tokens with respect to both spatial and temporal order, which we refer to as "full perm." and the results are shown in the second row of Figure 7. By contrast, our method permutes the target tokens only respect to spatial dimension while keeping the original temporal order. All of the permutation strategies show similar visual results on frames close to input view. The quality of "raster" strategy degrades significantly at later frames. This is because the images data with bi-directional context is applied to optimize a uni-directional model, which may fall into sub-optimal solutions. The "full perm." strategy also produces less quality results as the temporal order generation is also random. This means target views far from input view could be generated earlier than those close to the input view, thus failing to condition on the knowledge of previous views. Our full method presents the overall best visual results compared to other permutation strategies.

**Ablation on tokenization strategy.** We further conduct experiments on different choices of tokenizer. To validate the effectiveness of our video tokenizer, we apply the VQ image tokenizer to convert multi-view images into discrete tokens. To validate the temporal consistency of the generated sequence, we also report the FVD score except for the classic PSNR score and the quantitative results are shown in Table 3. our method achieves consistently superior performance across all

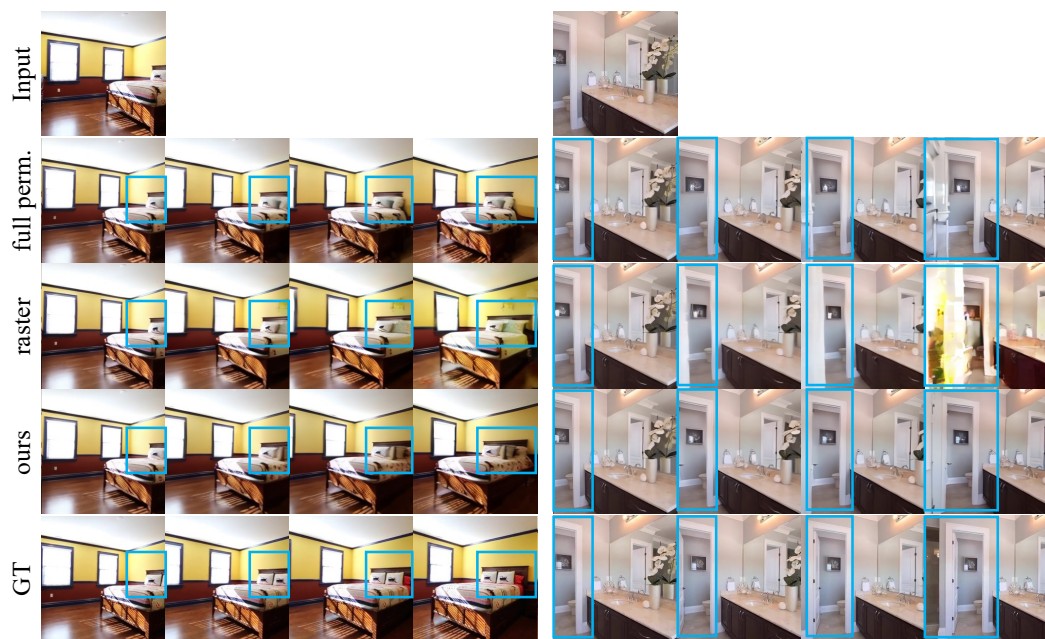

Figure 7: **Ablation Studies.** Visualization of different token permutations. In the figure, **full perm.** means to perform both spatial and temporal permutation of all the target tokens during training. **Raster** means to keep the original order of target tokens. Full permutation leads to incorrect geometry since later tokens may be generated first, whereas raster ordering causes visual distortions that grow as the generated frame becomes farther from the input view.

Table 2: **Ablation Studies.** We report metrics scores on different token permutation strategies. In the table, **raster** means to keep the original token order while **full perm.** means to randomly shuffle the token in both spatial and temporal dimension.

Table 3: **Ablation Studies.** We report metrics scores on different image tokenizers. In the table, **VQ** means to apply vector quantization image tokenization on the multi-view images. FVD score is evaluated to demonstrate the temporal consistency

| Method | PSNR ↑ | SSIM ↑ | LPIPS ↓ | FID ↓ |
|---|---|---|---|---|
| raster | 16.29 | 0.488 | 0.402 | 71.17 |
| full perm. | 18.76 | 0.532 | 0.315 | 62.58 |
| ours | **19.22** | **0.565** | **0.294** | **60.11** |

| Method | PSNR ↑ | SSIM ↑ | LPIPS ↓ | FVD ↓ |
|---|---|---|---|---|
| VQ | 15.69 | 0.437 | 0.498 | 137.68 |
| ours | **19.22** | **0.565** | **0.294** | **52.56** |

metrics, with the FVD score improving by approximately 62%. This indicates that the VQ image tokenizer fails to preserve temporal consistency, whereas the video tokenizer can effectively maintain.

## 5 DISCUSSION

To the best our knowledge, ARSS is the first work that uses causal autoregressive models to generate view consistent sequences with camera control from a single image. We use video finite scalar quantization to tokenize the multi-view images into 1-D discrete sequences and we design a camera autoencoder to map Plücker raymap into latent representations as 3D positional instruction tokens for visual tokens. The experimental results demonstrate that our method outperforms state-of-the-art methods leveraging diffusion models and transformers. The generation quality of ARSS is still limited by the quality of tokenizer. Although the current tokenizer is trained on tons of thousands of video datasets, it is hard to adapt to significant view changes thus would lead to the generaton of inferior discrete tokens. In the future, we will train a tokenizer that is designed for multi-view images. In addition, different from the current diffusion-based view synthesis method that mostly finetuned from pre-trained models, our method is trained from scratch using limited public datasets with relatively low resolution.

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

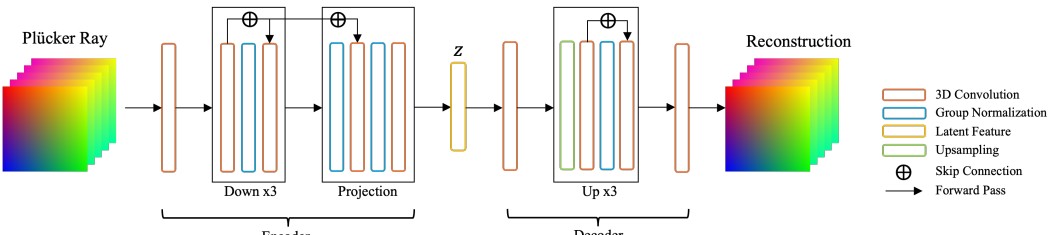

Figure 8: Architecture of our camera encoder

# A APPENDIX

## A.1 CAMERA AUTOENCODER ARCHITECTURE

The camera encoder maps the camera Plücker raymap with a compact 3D CNN whose strides are designed to match the dimension of visual tokens. The architecture is visualized in Figure 8. The encoder comprises a 3D CNN module followed by 3 downsample blocks. Each block uses residual 3D convolutions (He et al., 2016) with Group Normalization (Wu & He, 2018) and SiLU (Elfwing et al., 2018) activation. A bottleneck "post" projects the latent features to camera tokens. The decoder mostly mirrors the encoding but utilize upsampling module in each up blocks for reconstruction.

## A.2 ADDITIONAL IMPLEMENTATION DETAILS

**Classifier-free guidance (CFG).** To support CFG, during training, the input camera and visual tokens would be dropped with a probability 10%. During sampling time, the model would be called based on both conditional input tokens and unconditional tokens, which we denote us $x_c$ and $x_u$ for simplicity. The logits of the generated token at step $t$ would be modified as: $\text{logit}(x_t) = f_\theta(x_t|x_u) + \omega \cdot (f_\theta(x_t|x_c) - f_\theta(x_t|x_u))$, where $\omega$ is the guidance scale.

## A.3 BROADER RESEARCH IMPACT

Our proposed method aims at pioneering the research to integrate the generative paradigm in large language model into novel view synthesis task, which, as far as we know, is the first work in this research area. Our proposed method has the potential to bring multimodal generative model into a unified training and sampling paradigm. Our future work would focus on 1) designing more specialized tokenizer for multi-view images to further improve the generation quality, and 2) collecting more high-resolution multi-view image sequences to achieve more robust training.

## A.4 ADDITIONAL ABLATION STUDIES

We provide qualitative results of our ablation study in Section 4.3 in Figure 7. The *raster* order keeps the original spatial and temporal sequence, while *full perm.* shuffles tokens across both dimensions. Our method permutes only the spatial dimension while preserving temporal order. All strategies perform similarly on frames near the input view, but raster degrades at later frames due to the mismatch between bi-directional image context and uni-directional modeling, and full permutation produces artifacts as distant views may be generated before closer ones. In contrast, our approach achieves the best visual quality across the sequence.

All strategies perform similarly on frames near the input view, but raster degrades significantly at later frames due to misalignment between bi-directional image context and uni-directional modeling, and full permutation performs poorly because distant views may be generated before closer ones. In contrast, our method achieves the best overall visual quality across sequences.

## A.5 ADDITIONAL VISUALIZATIONS AND COMPARISONS

We provide more visualization results compared with other state-of-the-art method in Figure 9 and more sequence generation results from single view input in Figure 10 and Figure 11

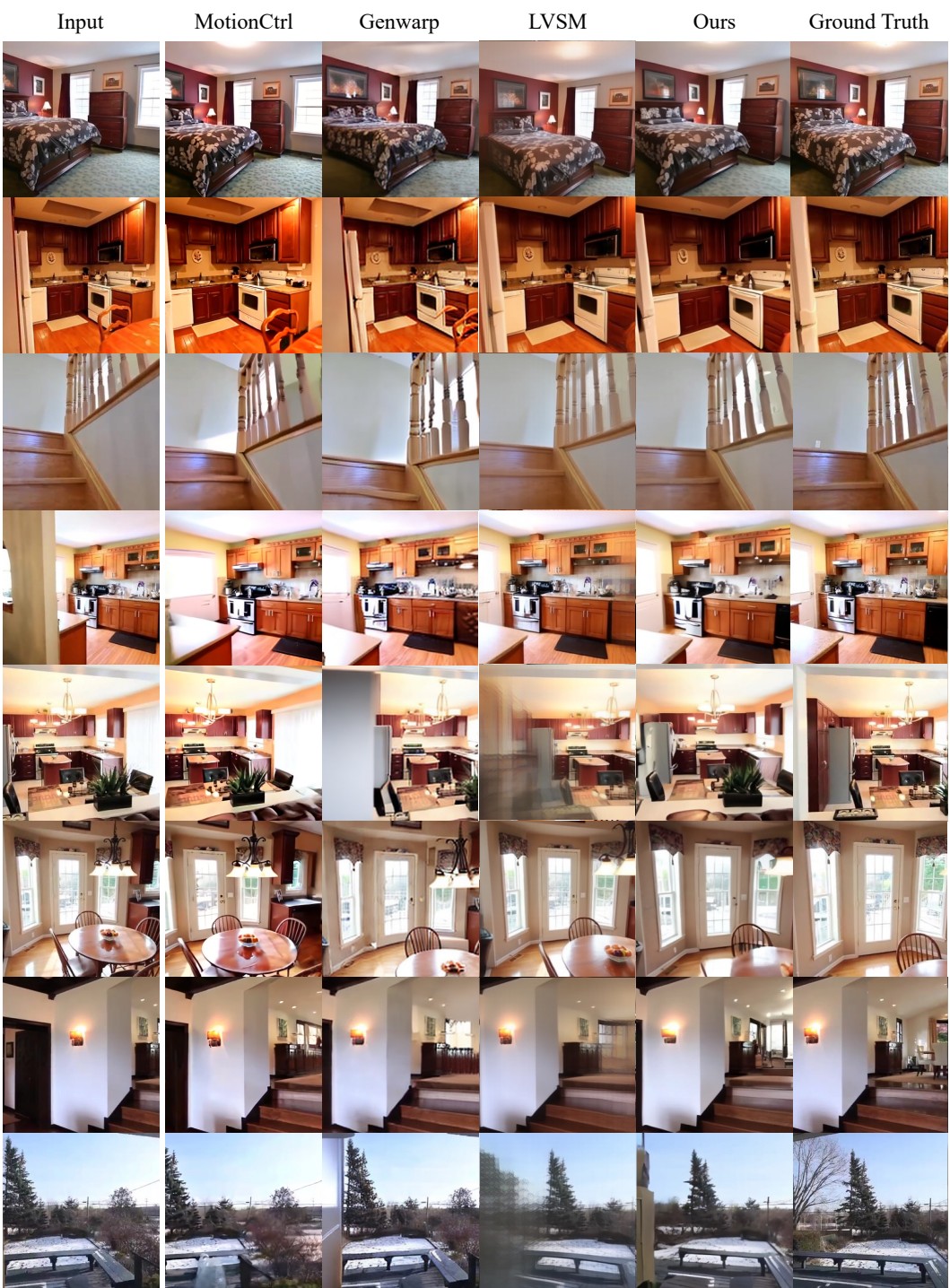

Input  MotionCtrl  Genwarp  LVSM  Ours  Ground Truth

Figure 9: **Qualitative Visualization.** Qualitative comparison between ARSS with other diffusion-based and feed-forward transformer-based methods on ReaEstate10K and ACID datasets. Diffusion-based methods such as SEVA and Genwarp often suffer from distortions and inaccurate camera pose alignment, while the feed-forward transformer-based LVSM produces results that are noticeably blurry along boundaries. In contrast, ARSS generates geometrically consistent and sharp views across diverse scenes.

Input                    Target

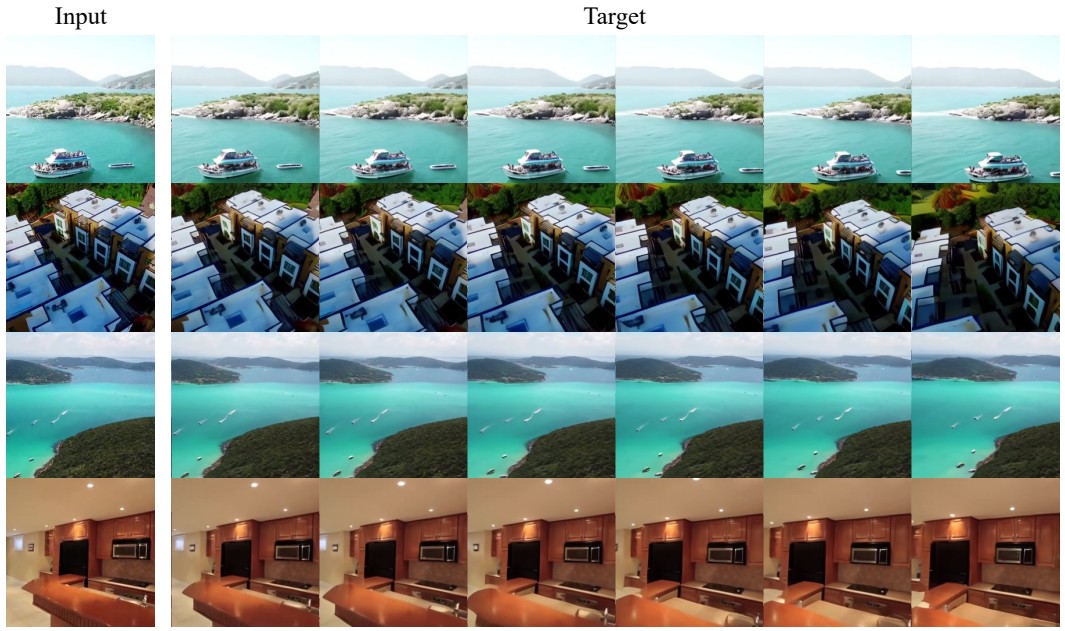

Figure 10: Qualitative visualization for multi-view sequence generation on RealEstate-10K (Zhou et al., 2018) and ACID (Liu et al., 2021) datasets

Input                    Target

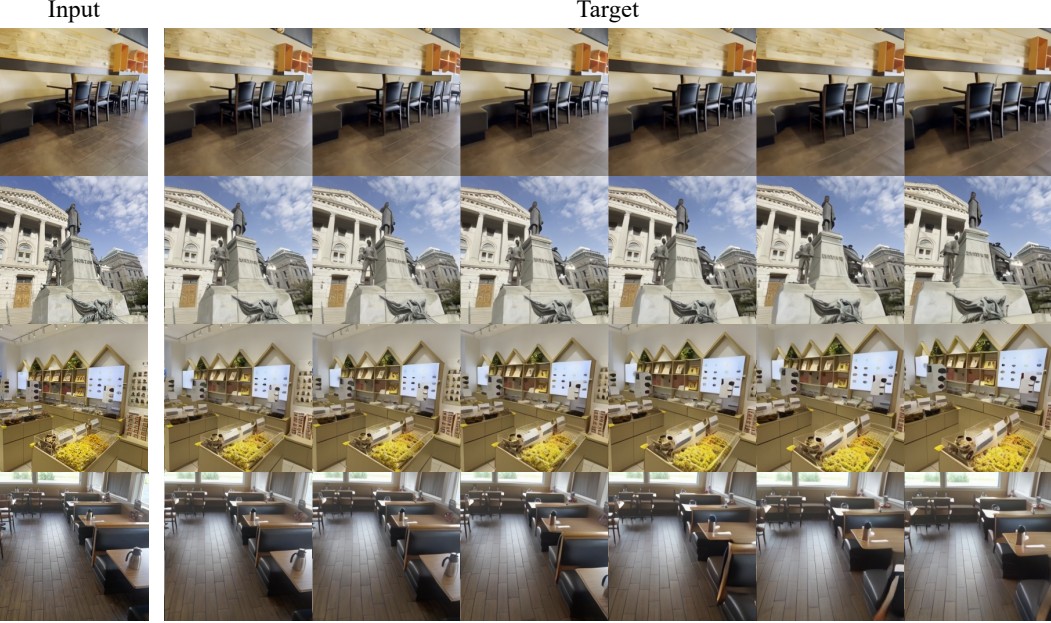

Figure 11: Qualitative visualization for zero-shot multi-view sequence generation on DL3DV benchmark (Ling et al., 2024) dataset

