# OpenReview forum: "ARSS: Taming Decoder-only Autoregressive Visual Generation for View Synthesis From Single View"
_ICLR.cc/2026/Conference — Submitted to ICLR 2026_

### Official Review · Reviewer_xUT2 · 2025-10-28

**Soundness:** 3
**Presentation:** 3
**Contribution:** 2
**Rating:** 4
**Confidence:** 5

**Summary:**

The paper presents an autoregressive transformer framework for novel-view synthesis conditioned on a predefined camera trajectory. Instead of an image-based tokenizer, the authors use a video-based tokenizer to reduce 3D inconsistencies. They also propose a Plücker-ray autoencoder to tokenize camera trajectories. After tokenizing both images and camera embeddings, a decoder-only transformer generates novel views via next-token prediction. Experiments show the proposed method achieves state-of-the-art results.

**Strengths:**

- The paper is well written and easy to follow.

- The design of the camera-embedding autoencoder is novel and highly useful and could be applied in future work.

- The results look strong and achieve state-of-the-art performance compared to prior work.

**Weaknesses:**

- The authors claim that “diffusion models have limited applicability to world modeling tasks, such as novel view generation from sparse inputs.” However, recent work (e.g., Gen3C, CVPR 2025) demonstrates promising results for novel-view synthesis from sparse inputs. Video-diffusion-based approaches also play an important role in world modeling and achieve competitive performance. Diffusion-based and feed-forward transformer-based methods each have their own strengths for these tasks, so the authors should avoid a biased presentation and instead acknowledge the complementary advantages and trade-offs.

- Using a video-based tokenizer is common in video-diffusion approaches for novel-view synthesis; it is not a novel contribution on its own.

-  Please compare with the latest related work, including transformer-based methods (e.g., Rayzer, ICCV 2025) and recent diffusion-based methods (e.g., Gen3C, ViewCrafter).

-  Novel-view synthesis: If I want a single specific target view, does your model require generating a predefined sequence, or can it directly produce a single image like image-based methods? As I understand it, your model generates views sequentially from input views. If so, generating a target view that is far from the inputs may require many steps and be time-consuming. Please clarify this limitation and discuss trade-offs compared to single-view generation methods.

-  Lack of visualization for temporal/video consistency: For sequences of novel-view images, please provide dynamic video renderings (continuous sequences), not only static frames. I could not find such videos in the supplementary material.

**Questions:**

- Line 316-317 in the paper, has some visualized artifacts.

- Missing some relative reference for novel-view synthetics:
       Zero-1-to-3: Zero-shot one image to 3d object
       RayZer: A Self-supervised Large View Synthesis Model
       Videomv: Consistent multi-view generation based on large video generative model
       ViewCrafter: Taming Video Diffusion Models for High-fidelity Novel View Synthesis



As noted in the Strengths and Weaknesses, I am currently inclined to give a borderline-reject score; I would be willing to raise this if the authors address the timing, baselines, and other main concerns.

**Details Of Ethics Concerns:**

There are not ethics issues.

---

> ### Author Response · Authors · 2025-11-20
> **Rebuttal to Reviewer xUT2**
>
> We sincerely thank reviewer for all the feedback and comments, below are our responses regarding each weaknesses and questions.
>
> $\textbf{w1.}$ We appreciate the reviewer’s concern and agree that our wording was inaccurate. We will revise the text to acknowledge that diffusion-based video and NVS models (e.g., Gen3C, ViewCrafter, VideoMV) have demonstrated strong performance for world modeling tasks. Our goal is not to argue that diffusion is unsuitable, but to present ARSS as another efficient approach that applies next-token prediction with explicit camera token guidance for sequence generation. Moreover, compared to diffusion-based NVS models, ARSS adopts a GPT-style next-token prediction objective with a single cross-entropy loss over discrete video tokens, without noise schedules or multi-step denoising objectives. In practice this leads to a simpler and more stable training recipe.
>
> $\textbf{w2.}$ We would like to clarify our intended claim. Our contribution is not to introduce video tokenizers per se, but to demonstrate that a discrete video tokenizer, when combined with (i) camera tokens and (ii) a decoder-only autoregressive transformer with spatial permutation, forms an effective architecture for novel view synthesis. To the best of our knowledge, existing diffusion-based NVS methods operate either directly in RGB space or in continuous latent spaces obtained from video/image VAEs, rather than in discrete video token space. In addition, discrete video tokenizers have not been commonly used in multi-view NVS with explicit camera conditioning.
>
> $\textbf{w3.}$ Here we include RayZer and ViewCrafter as two new baselines for a quantitative comparison. We also add the results in our main paper. The best score is highlighted with bold, whereas the second best score is highlighted with underline. For qualitative comparison, we encourage reviewer to take a look at our updated supplementary videos.
>
> **Table 1: Quantitative results on Re10K.**
>
> | Method      | PSNR↑ | SSIM↑ | LPIPS↓ | FID↓ | FVD↓
> |------------|------:|------:|-------:|-----:|-----:|
> | MotionCtrl | 16.17 | 0.609 | 0.438  | 59.73 | 63.58|
> | ViewCrafter|  12.67  | 0.399 | 0.490 | 121.25 | 108.99 |
> | RayZer     |   12.97 | 0.397 | 0.639| 324.23 | 130.23 |
> | LVSM       | 18.29 | 0.579 | $\underline{0.314}$  | 50.29 | $\underline{56.31}$ |
> | SEVA       | $\underline{18.73}$ | $\textbf{0.670}$ | 0.349  | $\textbf{46.98}$ | 57.56|
> | Ours       | $\textbf{19.02}$ | $\underline{0.624}$ | $\textbf{0.269}$  | $\underline{47.60}$ | $\textbf{50.51}$ |
>
> **Table 2: Quantitative results on ACID.**
>
> | Method      | PSNR↑ | SSIM↑ | LPIPS↓ | FID↓ | FVD↓ |
> |------------|------:|------:|-------:|-----:|-----:|
> | MotionCtrl | 19.36 | 0.626 | 0.405  | 63.93 | 66.30 |
> | ViewCrafter | 16.959 | 0.504 | 0.442 | 102.48| 104.85|
> | RayZer     | 12.635 | 0.384 | 0.6521| 303.75 | 138.62 |
> | LVSM       | 20.81 | 0.573 | $\underline{0.308}$  | $\underline{38.46}$ | 55.13 |
> | SEVA       | $\underline{21.77}$ | $\textbf{0.664}$ | 0.326  | $\textbf{33.16}$ | $\textbf{53.69}$ |
> | Ours       | $\textbf{21.93}$ | $\underline{0.623}$ | $\textbf{0.265}$  | 47.76 | $\underline{54.60}$ |
>
> $\textbf{w4.}$ ARSS can synthesize a single novel view without generating a long trajectory. In practice, to obtain one target image we specify a “trajectory” consisting of the source camera and the desired target camera only, and decode tokens for that target frame in a single pass.
>
> Regarding very large viewpoint changes, our current training setup mainly samples trajectories with moderate camera baselines, because the off-the-shelf video tokenizer tends to degrade when adjacent frames differ too drastically. As a result, extremely large source–target baselines can be challenging, which is a limitation shared by many NVS methods. We will state this limitation more explicitly and plan to explore multi-view images tokenizers that are trained to handle larger spatial coverage, which we expect to further improve single-view performance under extreme camera changes.
>
> $\textbf{w5.}$ We apologize for not including videos, we will add videos for baseline comparisons in the revised supplementary materials.
>
> $\textbf{q1.}$ Thanks for pointing out, we would modify in our revised version.
>
> $\textbf{q2.}$ Thanks for pointing out. We will include the methods in our "Related Works" section of the revised main paper.

---

> ### Comment · Reviewer_xUT2 · 2025-11-24
> **Official Comment by Reviewer xUT2**
>
> ***To authors:***
>
> I appreciate the authors' detailed response and the additional quantitative comparisons and video demos comparing their results to state-of-the-art methods. However, after reviewing the supplementary videos, the visual results do not convincingly support the claimed quantitative improvements. Consequently, I find it difficult to trust the reported results and therefore maintain my initial scores. Below are my concerns after reading the authors' response:
>
>
> ---
>
>
> ***Video demos in the supplementary material:***
>
> - For acid_2b36ad9e8171135d: the video is static — the background does not move.
> - For acid_2cd6bd8f20240228 and acid_26dae0963026e814: I do not see a significant improvement over LVSM; the results appear blurrier, not sharper.
> - For re10k_0a78dcb828c506f1: there are high-frequency artifacts on the stair railing compared to LVSM. Does this indicate the model struggles to reproduce fine-grained details?
>
> ***Comparison with SOTA***
>
> - The RayZer results are not convincing. How were you running the code, and why is RayZer worse than LVSM in your experiments?
> - For Gen3C, did you use the depth / point-cloud conditioning? I suspect you did not use point-cloud–conditioned novel-view generation.

---

### Official Review · Reviewer_r8gv · 2025-10-31

**Soundness:** 3
**Presentation:** 3
**Contribution:** 2
**Rating:** 4
**Confidence:** 4

**Summary:**

This paper introduces ARSS, a new framework for novel view synthesis (NVS) from a single input image. The core idea is to move away from diffusion-based models and instead use a GPT-style decoder-only autoregressive (AR) model. The system works by first encoding the input image and a target camera trajectory. It uses a pre-trained video tokenizer to convert image sequences into discrete tokens (to maintain temporal consistency) and a new camera autoencoder to turn camera paths into 3D positional guidance tokens. The model then autoregressively predicts the next visual token, conditioned on all previous tokens and the camera guidance. To help the uni-directional AR model handle 2D image data, the authors propose a hybrid token-shuffling strategy where the spatial order of tokens within a frame is randomized, but the temporal (frame-by-frame) order is kept fixed. The authors claim this method achieves results comparable to or better than state-of-the-art diffusion models on standard NVS benchmarks.

**Strengths:**

1. Token Permutation Strategy: The idea of only permuting the spatial order of tokens while maintaining the temporal sequence is a clever trick. It attempts to get the best of both worlds: adapting the AR model to the bi-directional nature of images without breaking the causal, frame-by-frame generation process.

2. Logical Components: The proposed architecture is well-reasoned. Using a video tokenizer to enforce temporal coherence and a dedicated camera encoder to inject 3D positional guidance makes intuitive sense.

**Weaknesses:**

I have several major concerns that, in my opinion, prevent the paper from being ready for acceptance.

1. Missing Video Results: This is the most significant weakness. The paper tackles a video generation task (synthesizing a sequence of new views) but provides only static images in the figures. The primary justification for using a video tokenizer is to improve temporal consistency, but this claim is impossible to verify without seeing the actual video outputs. Static images can hide critical failures like flickering, temporal drift, or object instability. The authors even note in the appendix that AR models suffer from "error accumulation," and video results are the only way to judge how severe this problem is.

2. Insufficient Evidence for Performance Claims: The paper claims to perform comparably to or better than state-of-the-art methods, but the evidence provided is not convincing. The paper relies on a very small number of selected static images in the figures (e.g., Fig 3, 4, 8). This is not sufficient to demonstrate the general efficiency, robustness, or consistency of the method. Given the claims, a much more comprehensive set of qualitative results is needed to fairly judge the model's performance against strong baselines, especially since the provided quantitative metrics (Table 1) are mixed and do not show a clear win (e.g., SSIM and FID are often worse).

3. Heavy Reliance on Pre-trained Tokenizer: The entire framework is critically dependent on the quality of the "VidTok" video tokenizer, which was not designed specifically for this task. The authors admit this in their limitations, stating that "The generation quality of ARSS is still limited by the quality of tokenizer." This makes the contribution feel less fundamental and more like an integration of existing tools, whose own limitations (e.g., handling large view changes) are inherited by the proposed method.

4. Unanalyzed Error Accumulation: The paper mentions the classic AR problem of error accumulation but doesn't analyze it. This is a key potential drawback compared to diffusion models that often generate an entire sequence in parallel. How does the image quality degrade as the camera path gets longer? A quantitative study on this (e.g., plotting PSNR vs. frame number) would be essential for a fair comparison, but it is missing.

5. Unverified Claims of Camera Pose Accuracy: The paper claims (e.g., in the caption for Figure 4) that baseline methods suffer from "incorrect camera pose alignment." However, it provides no quantitative or qualitative proof (like rendering a known 3D object or re-projecting points) to demonstrate that its own method accurately adheres to the provided camera trajectories. The visual results are not sufficient to verify this claim, especially when the camera autoencoder (Sec 3.2.2) is a core part of the method.

**Questions:**

1. How do you interpret the significant gap in FID scores on the ACID dataset, despite the competitive LPIPS scores? Does this point to a lack of diversity or a consistent "style" bias introduced by the AR model or tokenizer?

2. Regarding the camera autoencoder: Have you performed any ablation studies on its reconstruction quality? How much geometric precision is lost during this tokenization step, and how sensitive is the final output quality to this loss?

3. You mention error accumulation as a limitation. Have you quantitatively measured this? For example, how does the quality (e.g., PSNR/LPIPS) of the 10th generated frame compare to the 1st generated frame in a sequence?

---

> ### Author Response · Authors · 2025-11-20
> **Rebuttal to Reviewer r8gv**
>
> We sincerely thank reviewer for all the feedback and comments, below are our responses regarding each weaknesses and questions.
>
> $\textbf{w1.}$ We sincerely apologize for not including videos, we will add videos for baseline comparisons in the revised supplementary materials. For error accumulation analysis, we calculate per-frame PSNR, SSIM and LPIPS to compare the error accumulation comparisson between our method and baseline methods. Please refer to Figure 6. of revised main paper.
>
> $\textbf{w2.}$ We have significantly expanded our qualitative evaluation. In the supplementary material, we now provide (i) videos on our test datasets that show camera trajectories with side-by-side comparisons between our method and strong baselines (SEVA, LVSM, RayZer, ViewCrafter, MotionCtrl), and (ii) large image grids with many additional test scenes, each including multiple target views per scene. These results allow a much more comprehensive inspection of efficiency, robustness, and temporal/multi-view consistency than the few static examples in Figs. 3, 4, and 8 of the main paper. In addition, the new error accumulation analysis (see Figure 6 in main paper) and pose accuracy anaylysis (see below) jointly show that rather than suggesting a uniform win in metric score our proposed method can produce both photo-realistic and 3D consistent multi-view sequence.
>
> $\textbf{w3.}$ We agree that ARSS currently relies heavily on the pre-trained VidTok tokenizer, and we have mentioned this as a limitation. Our contribution is to show that, given an off-the-shelf video tokenizer, a GPT-style decoder-only AR model with camera tokens can give competitive NVS performance. We also provide the ablation study in the second part of Section 4.3 and compare the choice of video tokenizer and image tokenizer.
>
> $\textbf{w4.}$ As mentioned before, please see Figure 6 of the revised supplementary material for error accumulation analysis.
>
> $\textbf{w5.}$ For camera pose accuracy, we use the generated frames for scene reconstruction and pose estimation and compare the estimated pose against the ground truth camera pose. Here are both of the translation and rotation vector comparisons. The errors are measured in degree and the best scores are highlighted in bold.
> **Table 1: Quantitative results on Re10K.**
>
> | Method      | Rotation Error↓| Translation Error↓ |
> |------------|------:|------:|
> | MotionCtrl | 4.05 | 52.38 |
> | ViewCrafter| 5.27 | 40.50 |
> | RayZer     | 71.47 |  51.73 |
> | LVSM       | 2.46 | 35.14 |
> | SEVA       | 0.88 | 21.05 |
> | Ours       | $\textbf{0.68}$ | $\textbf{18.12}$ |
>
> **Table 2: Quantitative results on ACID.**
>
> | Method      | Rotation Error↓| Translation Error↓ |
> |------------|------:|------:|
> | MotionCtrl | 3.20 | 72.68 |
> | ViewCrafter| 4.42 | 52.87 |
> | RayZer     |  87.94 | 69.30 |
> | LVSM       | 2.13 | 41.56 |
> | SEVA       | 1.52 | 36.16 |
> | Ours       | $\textbf{1.30}$ | \$\textbf{25.59}$ |
>
> These quantitative analysis illustrate that our method can generate scenes with more accurate camera trajectories.
>
> $\textbf{q1.}$ We agree with the reviewer that there is a noticeable gap in FID on ACID, despite our competitive LPIPS and PSNR scores. Our interpretation is that this does not reflect a lack of diversity or severe mode collapse. FID measures distribution-level statistics of Inception features over the entire set of generated images, and is known to be sensitive to consistent changes in color tone, contrast, or texture, even when individual images are perceptually close to the ground truth. In our qualitative results on ACID (see supplemtary videos), ARSS often produces photo-realistic and 3D consistent views compared to other baseline methods against ground truth, which improves per-image perceptual similarity.
>
> $\textbf{q3.}$ Again, as mentioned before, please see Figure 6 of the revised supplementary material for error accumulation analysis.

---

> ### Comment · Reviewer_r8gv · 2025-11-24
>
> I thank the authors for their detailed response and for providing the additional quantitative comparisons and error accumulation analysis. However, after carefully reviewing the new data, specifically Figure 6 and the comparison with ViewCrafter, I have significant reservations regarding the fairness of the comparison and the accuracy of the reported baseline metrics. My concerns are as follows:
>
> **1. Unconvincing Error Accumulation Analysis (Figure 6)**
>
> I am not convinced by the error accumulation analysis presented in Figure 6.
>
> * **Sequence Length & Resolution:** You compare your method (generating 17 frames at 256x256) against baselines like ViewCrafter, which is designed to generate 25 frames at much higher resolutions (576x1024). **Crucially, a sequence of 17 (or even 25) frames is insufficient to meaningfully evaluate "error accumulation."** In Autoregressive models, significant drift typically manifests over much longer horizons. To genuinely validate claims of superior "long horizon" stability, the evaluation must be conducted on sequences significantly longer than 25 frames (eg. 200 frames). Claiming superior stability based on such a short, low resolution sequence compared to a full sequence diffusion model is inadequate and does not prove the method avoids error accumulation.
>
> * **Unrealistic Baseline Performance:** In Figure 6, the PSNR for baselines (ViewCrafter, MotionCtrl, RayZer) drops below 12 dB after roughly 10 to 14 frames. For a full sequence diffusion model, a PSNR < 12 implies a catastrophic failure (e.g., essentially noise or completely wrong content). This contradicts the visual performance typically observed in these models and their official evaluations. It strongly suggests that the baselines may not have been evaluated correctly (e.g., issues with coordinate system alignment, resolution scaling, or conditioning).
>
> **2. Discrepancy in Quantitative Results (Table 1 from Reviewer xUT2 and G3wz)**
>
> The quantitative results reported in the rebuttal contradict established benchmarks.
>
> * You report a PSNR of **12.67** and SSIM of **0.399** for ViewCrafter on RealEstate10K.
>
> * However, the official ViewCrafter paper and repository report significantly higher performance on similar benchmarks. Given that ViewCrafter generates longer sequences with higher fidelity, the extremely low scores reported here cast doubt on the validity of the implementation used for comparison. Unless the specific evaluation protocol (e.g., strict resizing to 256x256 without proper antialiasing, or specific test set deviations) is clarified, I cannot accept that your method outperforms these baselines by such a massive margin based on the provided table.
>
> **3. Lack of Explanation for Camera Pose Accuracy**
>
> Regarding the camera pose evaluation, you report a rotation error of **0.68** for your method versus **5.27** for ViewCrafter.
>
> * Both methods utilize Plücker Ray embeddings to encode camera conditions.
>
> * The core difference is the generative mechanism (AR vs. Diffusion). It is unclear *why* the AR paradigm, which inherently suffers from accumulated drift, would yield an order of magnitude improvement in pose alignment compared to a global diffusion process that considers the full sequence context which only generate 17 frames.
>
> * The paper does not provide a technical justification for this. Is it the specific quantization in the Camera Autoencoder? Without a logical explanation or an ablation of the Camera Autoencoder vs. standard embeddings, this performance gap appears anomalous.
>
> **Conclusion**
>
> While I appreciate the effort to add baselines, the reported performance for State of the Art methods (specifically PSNR < 12 for ViewCrafter) is inconsistent with the performance of the published paper. This raises concerns that the baselines are not being evaluated under optimal or standard conditions. Consequently, I cannot verify the claim that the proposed ARSS framework achieves SOTA performance.

---

### Official Review · Reviewer_5asS · 2025-11-01

**Soundness:** 3
**Presentation:** 3
**Contribution:** 3
**Rating:** 6
**Confidence:** 4

**Summary:**

This paper introduces ARSS, a  framework that uses a GPT-style, decoder-only autoregressive model to generate novel views from a single image conditioned on a predefined camera trajectory. Unlike previous work, ARSS employs a video tokenizer to convert continuous image sequences into discrete tokens and a camera encoder to translate camera trajectories into 3D positional guidance. To improve generation quality while retaining the autoregressive structure, ARSS includes an autoregressive transformer module that randomly permutes the spatial order of tokens while preserving their temporal order. Extensive experiments show that ARSS performs on par with or better than state-of-the-art diffusion-based view-synthesis methods.

**Strengths:**

1. The camera autoencoder is novel: it encodes camera trajectories into tokens that can be easily integrated into transformer architectures.

2. Video tokenization improves 3D consistency; visualizations show the method recovers higher-fidelity results than prior work.

3. The approach achieves state-of-the-art results—especially compared with diffusion-based view-synthesis methods—in both qualitative and quantitative evaluations.

4. The writing flows well and is reader-friendly.

**Weaknesses:**

The paper is novel and effective; the following points are offered as constructive suggestions for improvement.

1. Autoregressive framework: If I only want to generate a single novel view, does the model need to generate an entire camera trajectory to reach that view, or can it directly synthesize a single target image like LVSM? Please clarify the expected usage and efficiency for single-view generation.

2. 3D consistency: Which component contributes most to 3D consistency — the autoregressive generation scheme or the video tokenization? Please clarify their individual roles and relative importance.

3. Missing references: Please add comparisons or citations for recent related work, e.g., Rayzer (ICCV 2025), Cat4D (CVPR 2025), and Gen3C (CVPR 2025).

**Questions:**

None

---

> ### Author Response · Authors · 2025-11-20
> **Rebuttal to Reviewer 5asS**
>
> We sincerely thank reviewer for all the feedback and comments, below are our responses regarding each weaknesses.
>
> $\textbf{w1.}$ ARSS can synthesize a single novel view without generating a long trajectory. In practice, to obtain one target image we specify a “trajectory” consisting of the source camera and the desired target camera only, and decode tokens for that target frame in a single pass. The computational cost in this setting is comparable to LVSM, since the model only processes a short context and predicts one frame worth of tokens.
>
> Regarding very large viewpoint changes, our current training setup mainly samples trajectories with moderate camera baselines, because the off-the-shelf video tokenizer tends to degrade when adjacent frames differ too drastically. As a result, extremely large source–target baselines can be challenging, which is a limitation shared by many NVS methods. We will state this limitation more explicitly and plan to explore multi-view–aware video tokenizers that are trained to handle larger spatial coverage, which we expect to further improve single-view performance under extreme camera changes.
>
> $\textbf{w2.}$ Empirically, the video tokenizer is the primary driver for 3D consistency: when we replace it with a per-frame image tokenizer (VQ), the LPIPS/FVD metrics degrade (Tab. 3) noticeably and we observe more visible flicker across adjacent views, indicating that encoding spatio-temporal patterns jointly is crucial for stable geometry and appearance.
>
> At the same time, we believe that the autoregressive generation scheme is also important for 3D consistency along the camera trajectory. Unlike frame-wise baselines such as LVSM/SEVA that predict each target view independently from the source image, ARSS predicts tokens sequentially and allows every new token to attend to all previously generated tokens across space and time. Consistent with this interpretation, our per-frame PSNR/SSIM/LPIPS analysis (Figure 6 in revised main paper) shows that ARSS exhibits significantly slower degradation over frame index than non-autoregressive baselines, i.e., it accumulates less error over long trajectories.
>
>
> $\textbf{w3.}$ Thanks for pointing out, we will add these methods to our "Related Works" section of our revised main paper.

---

> ### Comment · Reviewer_5asS · 2025-11-24
> **Official Comment by 5asS**
>
> I have read all reviewers' comments, reviewed the updated supplementary videos, and examined the authors' quantitative results. Thank you to the authors for the detailed responses. I agree with Reviewers xUT2 and r8gv that the current evidence does not fully support the claims, and the visual demos do not show a clear improvement. I also share xUT2’s concern: why are RayZer’s results worse than LVSM? This is unclear. Moreover, It is confused whether the warped point clouds is used to condition for Gen3C? If not, the comparison experiments is unfair. If you use, why the results is so terrible?
>
> Accordingly, I will maintain my original score unless the authors address these issues.

---

### Official Review · Reviewer_G3wz · 2025-11-05

**Soundness:** 2
**Presentation:** 3
**Contribution:** 3
**Rating:** 2
**Confidence:** 3

**Summary:**

The authors propose an autoregressive framework (ARSS) for novel view synthesis: given a single input image and a specified camera trajectory, the model sequentially generates new views along that trajectory. They discretize images into visual tokens using a video tokenizer and introduce a camera trajectory encoder that maps the sequence of Plücker ray embeddings representing the trajectory into a set of latent camera tokens. These tokens provide 3D geometric guidance for the autoregressive decoder. The approach is compared primarily to diffusion-based view synthesis methods.

**Strengths:**

- The proposed method achieves better or comparable results with the compared baselines
- The proposed spatial permutation seems to improve the results compared to no or full permutation.
- Authors provide ablation justifying their design choices.

**Weaknesses:**

Over-generalization and inaccurate claims about related work and novelty:
- In the abstract, the authors claim that diffusion models are limited for novel view synthesis (NVS) due to their “non-causal” nature. This statement is not well-supported and oversimplifies the role of causality in diffusion-based generation.
- Throughout the paper, “autoregressive prediction” is used narrowly to refer only to the proposed decoder-only, next-token prediction scheme. The related work section overlooks a large body of causal “next-frame” or “next-view” approaches for video and NVS tasks (e.g., CausVid, ViewCrafter). This results in an incomplete and somewhat misleading positioning of the proposed method’s novelty.

Limited qualitative evidence of temporal or view consistency:
- It is difficult to assess the consistency of the proposed approach from the static figures provided. Including video or interactive visualizations (as supplementary materials) would substantially strengthen the evaluation of temporal and multi-view coherence.

Baselines:
-  As the method treats NVS as a next-view prediction task, it would have been helpful to include comparisons with video-generation models (e.g. CausVid, VideoCrafter 2) or NVS methods built on video-generation frameworks (e.g. ViewCrafter, SplatDiff, )

**Questions:**

- Could author provide more details on what per-frame mean in this sentence: "Since Genwarp (Seo et al.,
2024) and LVSM (Jin et al., 2024) are not designed for sequence generation, we adapt them by performing per-frame generation". Is the generation of each frame still conditioned on previous frames?

---

> ### Author Response · Authors · 2025-11-20
> **Rebuttal to Reviewer G2wz**
>
> We sincerely thank reviewer for all the feedback and comments, below are our responses regarding each weaknesses and questions.
>
> $\textbf{w1.}$ We thank the reviewer for pointing out that our abstract’s wording is too strong and oversimplifies the role of causality in diffusion-based generation. We agree that diffusion models can be used in causal or sequential ways for video and NVS (e.g., next-frame/next-view formulations), and we do not intend to claim that diffusion is fundamentally unsuitable for NVS. Our intended point is more specific: many existing diffusion-based NVS methods generate all target views jointly in a multi-step denoising process. This full sequence formulation makes it hard to apply the previously generated information for future generation guidance. In addition, compared to diffusion-based NVS method, ARSS adopts a GPT-style next-token prediction objective with a single cross-entropy loss over discrete video tokens, without noise schedules or multi-step denoising objectives. In practice this leads to a simpler and more stable training recipe.
>
> $\textbf{w2.}$ We fully agree that several recent diffusion-based methods for video and NVS (e.g., CausVid, ViewCrafter) are also causal in the sense that they generate frames or views sequentially. We refer “autoregressive prediction” to “next-token prediction” which is different from diffusion model or AR-based diffusion model. We will update from “autoregressive prediction” to “next-token prediction” in our revised version of the paper. We will also include in the related work a discussion on other works that combine diffusion model with autoregressive/iterative generation.
>
> $\textbf{w3.}$ We would add video visualizations in supplementary materials and add FVD in our main paper.
>
> $\textbf{w4.}$ Here are the quantitative results adding ViewCrafter and RayZer as baseline methods for comparison. The best score is highlighted with bold, whereas the second best score is highlighted with underline. For qualitative comparison, we encourage reviewer to take a look at our updated supplementary videos.
>
> **Table 1: Quantitative results on Re10K.**
>
> | Method      | PSNR↑ | SSIM↑ | LPIPS↓ | FID↓ | FVD↓
> |------------|------:|------:|-------:|-----:|-----:|
> | MotionCtrl | 16.17 | 0.609 | 0.438  | 59.73 | 63.58|
> | ViewCrafter|  12.67  | 0.399 | 0.490 | 121.25 | 108.99 |
> | RayZer     |   12.97 | 0.397 | 0.639| 324.23 | 130.23 |
> | LVSM       | 18.29 | 0.579 | $\underline{0.314}$  | 50.29 | $\underline{56.31}$ |
> | SEVA       | $\underline{18.73}$ | $\textbf{0.670}$ | 0.349  | $\textbf{46.98}$ | 57.56|
> | Ours       | $\textbf{19.02}$ | $\underline{0.624}$ | $\textbf{0.269}$  | $\underline{47.60}$ | $\textbf{50.51}$ |
>
> **Table 2: Quantitative results on ACID.**
>
> | Method      | PSNR↑ | SSIM↑ | LPIPS↓ | FID↓ | FVD↓ |
> |------------|------:|------:|-------:|-----:|-----:|
> | MotionCtrl | 19.36 | 0.626 | 0.405  | 63.93 | 66.30 |
> | ViewCrafter | 16.959 | 0.504 | 0.442 | 102.48| 104.85|
> | RayZer     | 12.635 | 0.384 | 0.6521| 303.75 | 138.62 |
> | LVSM       | 20.81 | 0.573 | $\underline{0.308}$  | $\underline{38.46}$ | 55.13 |
> | SEVA       | $\underline{21.77}$ | $\textbf{0.664}$ | 0.326  | $\textbf{33.16}$ | $\textbf{53.69}$ |
> | Ours       | $\textbf{21.93}$ | $\underline{0.623}$ | $\textbf{0.265}$  | 47.76 | $\underline{54.60}$ |
>
> For VideoCrafter 2 and CausVid, we would like to clarify that both methods target a different problem setting from ours. ARSS is designed for camera-pose–conditioned novel view synthesis. given a source image and an explicit sequence of camera poses, it must generate views that follow this trajectory and remain consistent with the underlying 3D scene. In contrast, the publicly available versions of VideoCrafter and CausVid are generic video generation models (primarily text-to-video, and in some cases image-to-video) that do not accept camera intrinsics/extrinsics as input and do not enforce adherence to a specified camera path. For SplatDiff, no public code is available for us to evaluate.
>
> $\textbf{q1.}$ We agree that the wording in this sentence is somwhat misleading. Both LVSM and Genwarp are per-frame generation, which means given source camera and image and target camera, these method directly generate the target view without conditioned on the previously generated frames. We will correct the wording for this sentence.

---

### Author Response · Authors · 2025-11-20
**Thanks all reviewers for all the valuable comments**

We thank all reviewers for their valuable comments and suggestions. In response, we have substantially revised both the main paper and the supplementary material.
- We now provide supplementary videos showing qualitative comparisons between our method and all baselines on the test datasets, which allow reviewers to directly inspect temporal and multi-view consistency.
- In the main paper, we introduce an explicit error-accumulation analysis with per-frame PSNR/SSIM/LPIPS curves over long trajectories.
- We further extend the quantitative comparison table to include two additional baselines, ViewCrafter and RayZer and add FVD scores to compare temporal consistency.
- We carefully revise the wording throughout the introduction, abstract, and related work to better position our contributions relative to diffusion-based and causal NVS methods, addressing all concerns about over-claims and terminology.
- In the related work section, we additionally cite and discuss the methods highlighted by the reviewers (e.g., CausVid, VideoCrafter, Gen3C, RayZer, ViewCrafter).

---

### Meta-Review · Area_Chair_kZcm · 2026-01-07

**Summary:**

The paper presents ARSS, a novel autoregressive framework for novel view synthesis (NVS) from a single input image, utilizing a GPT-style decoder model and incorporating a video tokenizer to enforce 3D consistency. However, there are significant unresolved concerns about the validity of the performance claims, particularly regarding the comparison with other methods and the temporal consistency of the generated views. The lack of video results and the reliance on a pre-trained tokenizer weaken the case for ARSS as a fundamentally novel approach.

The paper presents promising ideas but requires further validation and refinement before it can be accepted. Specifically, more robust baseline comparisons, clearer justification of performance discrepancies, and improved evidence of temporal consistency would be necessary. I recommend rejection at this stage, with the possibility of reconsideration if major revisions are made.

**Reviewer Concerns:**

Several reviewers pointed out that the novelty of the video tokenizer was overstated. Using video tokenizers for NVS is not new, and the paper should clarify that the key innovation lies in the combination of video tokenization with an autoregressive framework and camera conditioning.

There is also a notable discrepancy between the reported results and established baselines. Some reviewers questioned the validity of the results, especially in terms of PSNR scores for ViewCrafter and RayZer. The authors have provided updated comparisons and additional videos, but doubts remain regarding the fairness and accuracy of the baseline evaluations.

Moreover, despite the claim that ARSS improves 3D consistency, reviewers found that the lack of video results made it difficult to verify this. The suggestion to include more dynamic visualizations (e.g., videos) was repeatedly raised.

Lastly, concerns were raised regarding the lack of detailed error accumulation analysis, as well as the unexplained gap in camera pose accuracy between ARSS and diffusion models.

**Reviewer Scores:**

Given the feedback, especially from Reviewers G3wz and xUT2, the paper may struggle to reach acceptance, with the lack of clear improvement in baseline comparisons and temporal consistency weighing heavily.

---

### Decision · Program_Chairs · 2026-01-26

Reject